# Diagnostic Concordance between Research and Clinical-Based Assessments of Psychiatric Comorbidity in Anorexia Nervosa

**DOI:** 10.3390/jcm11247419

**Published:** 2022-12-14

**Authors:** Paola Longo, Federica Toppino, Matteo Martini, Matteo Panero, Carlotta De Bacco, Enrica Marzola, Giovanni Abbate-Daga

**Affiliations:** Eating Disorders Center for Treatment and Research, Department of Neuroscience, University of Turin, Via Cherasco 11, 10126 Turin, Italy

**Keywords:** anorexia nervosa, psychiatric comorbidity, inter-rater reliability, SCID-5, diagnostic assessment

## Abstract

The literature has reported poor concordance in the assessment of psychiatric conditions, and inhomogeneity in the prevalence of psychiatric comorbidities in Anorexia Nervosa (AN). We aimed to investigate concordance level between clinicians’ and researchers’ diagnoses of psychiatric comorbidity in AN and differences in eating and general psychopathology between patients with and without psychiatric comorbidity assessed by clinicians versus researchers. A clinical psychiatrist interviewed 122 patients with AN; then a researcher administered the Structured and Clinical Interview for DSM-5 (SCID-5). Participants completed the Eating Disorder Examination Questionnaire (EDE-Q), the State-Trait Anxiety Inventory (STAI), and the Beck Depression Inventory (BDI). The agreement between clinicians and researchers was poor for all diagnoses but obsessive-compulsive disorder and substance use disorder. Patients with comorbid disorders diagnosed by researchers reported more severe eating and general psychopathology than those without SCID-comorbidity. The differences between patients with and without comorbidities assessed by a clinician were smaller. Two approaches to psychiatry comorbidity assessment emerged: SCID-5 diagnoses yield a precise and rigorous assessment, while clinicians tend to consider some symptoms as secondary to the eating disorder rather than as part of another psychiatric condition, seeing the clinical picture as a whole. Overall, the study highlights the importance of carefully assessing comorbidity in AN.

## 1. Introduction

Anorexia nervosa (AN) is a severe mental disorder characterized by high mortality and psychiatric comorbidity, mostly concerning anxiety, depressive, and personality disorders [1,2]. Psychiatric comorbidity should be investigated with caution since it has been authoritatively suggested that it could be a by-product of current diagnostic systems [3]. Nevertheless, it was estimated that comorbidity in AN is an issue related to all illness phases and that patients are likely to gain other psychiatric diagnoses, also, after having recovered from AN [4,5]. Moreover, psychiatric comorbidities have a great impact on the clinical presentation of AN, influencing the prognosis [6] and the hospitalization course [7] and, importantly, frequently guiding clinicians’ choices concerning pharmacotherapy. More specifically, different authors showed that comorbidity could exacerbate the severity of eating-related symptoms, their chronicity, and their treatment resistance, leading also to frequent post-treatment relapses and a higher instability of the eating disorder (ED) diagnosis [8,9,10].

Despite the aforementioned importance of comorbid disorders in AN, the prevalence of comorbidities in AN is far from being conclusive. In fact, a wide range of prevalence is usually reported for each condition. In this context, according to Ulfvebrand and colleagues [2], the most common comorbidities in EDs are mood disorders, anxiety disorders, and substance use disorders (SUD), with generalized anxiety having the highest prevalence [2]. In contrast, another report described generalized anxiety as less common compared to other anxiety disorders [11]. Moreover, the lifetime prevalence of post-traumatic stress disorder (PTSD) in EDs has been reported as ranging from 9% to 24% [12], and, despite the high presence of traumatic events in AN [13], this disorder is often under-assessed [12,14]. Anxiety disorders yielded contrasting results too: the percentage of patients with AN and a punctual diagnosis of generalized anxiety ranges from 2% to 24% [2,15], while current comorbid panic disorder was diagnosed with a prevalence between 4% and 11.7% [2,11,15]. The same trend was observed for mood disorders, whose lifetime prevalence is reported to vary between 20% and 98% [8], and for obsessive-compulsive disorder (OCD) which was particularly studied in AN, especially in light of those articles reporting shared genetic and psychological features with AN [16,17]. However, in this case, as well, values range from 5% to 66% [11,15,18] considering both lifetime and current diagnoses.

Some methodological issues could be considered when analyzing such differences. On one hand, such variability could be due to changes in classification systems over time, pertaining to both AN and other disorders (i.e., changes in OCD between DSM-IV-TR and DSM-5 [19]). On the other hand, it could be also due to the different instruments and methods used to perform these assessments. Moreover, different kinds of samples recruited in the studies could influence the results on comorbidity. Indeed, samples could be different in the distribution of AN subtypes, and also the inclusion of male participants could make the difference, since a study reported that males are more likely to develop certain psychiatric comorbidities (e.g., substance use and obsessive behaviors) compared to female patients [2].

Additionally, there are multiple reasons for thinking that the picture of comorbidity could be even more complicated in the field of AN. In fact, it has been known since the 1850s that emaciation in AN could per se entail serious organic consequences, potentially encompassing other conditions like depressive symptoms [20]. Moreover, depression and anxiety could be the consequences of microbiome alterations caused by starvation and malnourishment [21], and authors have described obsessive-compulsive symptoms in individuals without an ED but in a state of semi-starvation [20,22]. Relatedly, the onset of comorbid conditions remains frequently unclear, also because sometimes patients have a hard time recollecting their clinical history [13].

Earlier studies reported a dismal lack of reliability in the assessment of psychiatric conditions [23,24], while more recent research showed more encouraging data [25]. For instance, a systematic review by Rettew et al. (2009) showed that the diagnostic agreement between standardized diagnostic interviews and clinical evaluations is highly heterogeneous based on the disorder assessed, and ranges from low to moderate accordance for the majority of diagnoses explored [26]. However, to the best of our knowledge, no studies have investigated the concordance between clinicians and researchers in the assessment of comorbid mental disorders in AN.

Prompted by the aforementioned methodological weaknesses and variability reported in the field, we designed this study to clarify the concordance between research and clinical-based assessment of mental disorders in AN; therefore, we compared psychiatric assessments performed by either clinicians or researchers with extensive training in the field of eating disorders to (a) assess the concordance levels between clinician- and researcher-based diagnosis of psychiatric comorbidity in AN; and (b) investigate any potential differences in eating and general psychopathology between groups of patients with psychiatric comorbidity as assessed by clinicians versus researchers.

## 2. Materials and Methods

### 2.1. Participants

The recruitment of participants took place between January 2018 and September 2021 and involved 140 patients voluntarily admitted to the Eating Disorders Centre of the Città della Salute e della Scienza hospital of the University of Turin, Italy. Of all contacted patients, 5 refused to participate in the study, 7 did not meet the inclusion criteria, and 6 did not provide a complete assessment. The present study’s final sample thus included 122 patients with AN. Patients were recruited from all the settings at the center: the vast majority of them (*n* = 107) were hospitalized, 10 of them were partially hospitalized in the day hospital, and 7 were tested during an outpatient visit.

Participants were involved in the study according to the following inclusion criteria:a.Diagnosis of AN according to the Structured and Clinical Interview for the DSM-5 [27];b.Age > 18 years. The following exclusion criteria were applied: (a) medical diagnoses (e.g., diabetes or neurological diseases); (b) anamnesis of cranial trauma with loss of consciousness.

All participants provided written informed consent according to the Ethical Committee of the University of Turin, which approved the present study under the registration number CS2/840.

### 2.2. Procedure

All patients were interviewed upon hospital admission by a psychiatrist with extensive training in the EDs field to collect clinical and demographic data and to confirm the diagnosis of AN. At the same time point, trained nurses measured patients’ height and weight to calculate their Body Mass Index (BMI).

As regards the assessment of psychiatric comorbidity, two experienced psychiatrists (two of the four psychiatrists in the clinical team) assessed for the presence of additional psychiatric diagnoses, performing one or more additional interviews in the first 4 days of hospitalization. Blind to the clinicians’ diagnostic procedure, 2 members of the research team, namely 1 clinical psychologist and 1 psychiatrist, administered the SCID -5 during the first days of hospitalization.

Then, patients completed a battery of self-reported questionnaires.

### 2.3. Materials

The Structured Clinical Interview for DSM-5 [27]: This is a standardized diagnostic interview to assess the presence of mental disorders based on DSM-5 criteria. Usually, it takes about 45–90 min to administer the interview [28]. The diagnoses included in the SCID-5 are psychotic spectrum disorders, bipolar disorders, depressive disorders, substance-related disorders, anxiety disorders, obsessive-compulsive disorders, post-traumatic stress disorder, attention-deficit/hyperactivity disorder, adjustment disorder, and other disorders. For this study, we took into account the following diagnoses: bipolar and depressive disorders (i.e., bipolar I, bipolar II, major depression disorder, persistent depressive disorder), substance-related disorders, anxiety disorders (i.e., panic disorder, social anxiety, generalized anxiety, agoraphobia), obsessive-compulsive disorder, and post-traumatic stress disorder.

The Eating Disorder Examination Questionnaire (EDE-Q [29]): The questionnaire investigates the frequency of ED-typical behaviors and thoughts during the last 28 days. It consists of 28 items, and the result includes four subscales (i.e., dietary restraint, eating concerns, weight concerns, and shape concerns) and a global score. The internal consistency is good, as shown by a Cronbach alpha value in our sample of 0.95.

The State-Trait Anxiety Inventory (STAI [30]): This tool assesses levels of anxiety with 40 items with answers ranging on a scale from 1 (never) to 4 (always). The questions are divided into two groups: 20 questions investigate the state anxiety (i.e., level of anxiety experienced at that moment), while the other 20 items ask for the trait anxiety (i.e., stable level of anxiety). In our sample, Cronbach’s alpha value was 0.60.

The Beck Depression Inventory (BDI [31]): It measures the presence and the severity of depressive symptoms. Participants are asked to answer 21 items. Based on the global score, the symptoms could be considered as low when the global score is between 0 and 4 and moderate when the global score is between 5 and 15. Finally, scores above 16 are an index of severe depressive symptomatology. This tool has a high internal consistency with a Cronbach’s alpha value of 0.89 in our sample.

### 2.4. Statistical Analysis

The SPSS 27.0 statistical software package (IBM SPSS Statistics for Windows, Version 27.0. Armonk, NY, USA: IBM Corp) was used for data analysis. To calculate the concordance between clinicians’ and researchers’ diagnoses, Cohen’s k was calculated. Cohen’s k values range between −1 and 1, where −1 indicates a complete disagreement between the raters, 0 suggests that the agreement can be attributed to the case, and 1 reflects a perfect agreement. More precisely, the concordance is considered poor for Cohen’s k values below 0, slight when Cohen’s k values are between 0 and 0.2, fair for values between 0.2 and 0.4, moderate when Cohen’s k is between 0.4 and 0.6, substantial for values between 0.6 and 0.8, and almost perfect for Cohen’s k between 0.8 and 1 [32].

Then, to investigate the differences in continuous variables between two groups, an independent sample t-test was conducted, while the Mann-Whitney test was used when the distribution of the groups was not normal. To compare groups based on categorical variables, we adopted the Exact Fisher’s test.

## 3. Results

The total sample consisted of 122 patients with AN, all Caucasian. Female patients represented the majority of the sample (98.4%). As regards AN subtypes, 41 inpatients were diagnosed with binge-purging AN (BP-AN; 33.6%) and 81 (66.4%) with restricter AN (R-AN). As regards treatment, 66 (54.1%) patients were under psychopharmacological treatment. In particular, 39 patients (32%) took benzodiazepines, 37 (30.3%) antipsychotics, 48 (39.3%) new-generation antidepressants, and 4 (3.3%) mood stabilizers.

### 3.1. Concordance between Clinical and SCID-5 Diagnosis in the Total Sample of Patients with AN

Concordance between comorbidities diagnosed by clinicians and those diagnosed by researchers with SCID-5 was overall low, as reflected by the Cohen’s k values suggesting poor inter-rater agreement for all the considered diagnoses but OCD, SUD, and mood disorders taken together (i.e., major depressive disorder, persistent depressive disorder, bipolar I and II disorders; Table 1).

### 3.2. Differences between Patients with and without Comorbidity Diagnosed with SCID-5

Comparing patients with and without at least one comorbidity assessed by SCID-5, no significant differences were found in the clinical variables. On the other hand, higher scores in all subscales of the EDE-Q, in trait and state anxiety, as measured by the STAI, and in depression severity according to BDI were observed in patients with SCID-5 comorbidity compared to those without it (Table 2). Finally, no significant difference was shown between patients with and without SCID-5 comorbidity in the distribution of AN subtypes (R-AN with comorbidity 63.9%; BP-AN with comorbidity 81.6%; Exact Fisher’s test *p* = 0.058), and in the frequency of patients taking psychiatric drugs (patients with comorbidity taking psychiatric drugs 68%; patients without comorbidity taking psychiatric drugs 50%; Exact Fisher’s test *p* = 0.068).

### 3.3. Differences between Patients with and without Comorbidity Diagnosed by Clinicians

No significant differences in clinical variables were found between patients with and without comorbidity diagnosed by clinicians. On the other hand, higher levels of eating concerns, state anxiety, and depression were observed in patients with clinical comorbidity compared to those without it (Table 3). No differences were found between the two groups in the frequency of AN subtypes (R-AN with comorbidity 41%; BP-AN with comorbidity 28.9%; Exact Fisher’s test *p* = 0.230) and in the frequency of patients taking psychiatric drugs (patients with comorbidity taking psychiatric drugs 66.7%; patients without comorbidity taking psychiatric drugs 60.6%; Exact Fisher’s test *p* = 0.342).

## 4. Discussion

Two main findings emerged from the present study: First, the concordance between clinicians and researchers in the assessment of psychiatric comorbidity in AN resulted as poor overall; a fair agreement was found for OCD and moderate for SUD. Second, the SCID-based diagnostic assessments yielded a more fine-grained evaluation of patients: those with a comorbid disorder reported more severe indices of eating and general psychopathology.

The first finding is in accordance with the metanalysis by Rettew and colleagues (2009), who found low to moderate agreement between diagnoses made with standardized diagnostic interviews and those reached through clinical evaluation [26]. Relatedly, each unipolar and bipolar mood disorder showed poor agreement between the raters. This result is in line with Thomas et colleagues (2010), who found slight accordance between clinicians’ and researchers’ assessments of major depressive disorder [25]. Despite the low agreement, the percentages of patients diagnosed with a unipolar mood disorder by both clinicians and researchers are in line with previous studies by Ulfvebrand and colleagues [2] and Himmerich et al. [33], but higher compared to the ones found by Mohammadi and coworkers [34]. It is noteworthy that, when mood disorders were taken together, the agreement reached a fair level. This could suggest that clinicians and researchers are discreetly in agreement on the presence of a mood disorder, but their accordance fails regarding the specific diagnosis.

Regarding anxiety disorders, the agreement between the raters was poor for all diagnoses, with the number of diagnoses made by the clinicians much less high compared to the number of SCID-5 diagnoses performed by the researchers. Overall, the percentages of patients with comorbid anxiety disorders as assessed by the researchers are in line with the literature, while the rate of anxiety-related diagnoses by the clinicians is lower compared to previous studies [2,15,35].

As regards OCD and SUD, the raters reached, respectively, fair and moderate agreement, even though the prevalence of the two diagnoses in this study was lower compared to previous data [33,34,35]. Relatedly, our sample included just patients with AN, especially of the restricter subtype, while the literature reports that patients suffering from the bulimic variant of AN are more likely to develop comorbidities, especially OCD [18].

The data together are partially in line with Mitchell et al. (2010), who compared diagnoses made by researchers with SCID-5 and those made by clinicians through unstandardized interviews with patients candidate for bariatric surgery. They found, indeed, low congruence between the two assessments for all explored diagnoses, including SUD [36]. However, they recruited a very different sample, and thus comparisons must be made with caution.

All of these data together led us to speculate that the poor agreement between the two assessments regarding mood and anxiety disorders could depend on different factors. First, the use of a structured interview can enhance precision and rigor in making categorical diagnoses; clinicians, indeed, often evaluate the situation as a whole without assessing the presence of all the required diagnostic criteria [25]. This approach may have some practical advantages (e.g., focusing on the most severe symptoms that required hospitalization), but it underlines how the so-called “clinical wise” can have limits. Second, it is known that depressive and anxious symptoms are so intertwined with the AN itself—particularly with the body image disturbance [37]—that the clinicians might consider such symptoms as dependent and consistent with the EDs, so that they are not considered as comorbidities. In other words, clinicians often consider depression and anxiety as secondary to the eating disorder itself. Clinicians, therefore, could be less prone to making pure comorbid diagnoses, while researchers using SCID-5 are more likely to separate the anxious and mood-related symptoms from the eating-related ones, coming thus to an independent diagnosis. This argument is related to the debate regarding pure or spurious comorbidity in eating disorders [37]. It is difficult to distinguish between a pure comorbid diagnosis (i.e., independent of the eating-related pathology) and a spurious comorbidity, a consequence of the eating disorder, and to establish what the main diagnosis is. On one hand, a more integrated hierarchical dimensional model of eating disorders is superior to DSM-5 diagnoses in characterizing a more appropriate clinical picture [38]; on the other, several studies have observed that the onset of anxiety disorders is often previous to the eating disorders [15], and other authors found shared genetic factors influencing the vulnerability to anxiety, depression, and eating disorders [35]: thus, patients who suffer from one of these disorders are also likely to develop the others. Longitudinal data could help to disentangle this issue. Regarding depression, many longitudinal studies are available describing depression and eating pathology as concurrent risk factors [39]; other studies showed major depression disorder in childhood as risk factors for eating disorders [40] and the absence of depression as a positive prognostic factor [41]. However, longitudinal studies specific to AN are still lacking. Lastly, considering that the majority of patients in this study were recruited in a hospital setting, the very acute phase of the illness could lead the clinician to consider some symptoms as sequelae of starvation. Future research on outpatients could clarify this point.

In contrast, OCD and SUD require that the symptoms are not related to food or weight, so these disorders are more easily detachable from the eating disorder. OCD and SUD symptoms, indeed, are particularly recognizable in specific behaviors and thus more detectable than symptoms related to anxiety and depression.

The PTSD data deserve a particular discussion. The agreement between the two raters on this diagnosis was not calculable, since the clinicians did not make any diagnosis of PTSD; meanwhile, the SCID-5 PTSD diagnosis percentage in the present study was higher than that in previous studies [12]. Several hypotheses could be proposed for the clinicians’ underestimation of PTSD: Post-traumatic symptoms, as well as the occurrence of the trauma, are often difficult for patients to report, especially because of feelings of guilt and shame [13]; SCID-5 provides a dedicated area for the assessment of PTSD, so asking directly to the patients about their traumatic history could encourage them to report it. Moreover, it is important to note that the participants in the present study were inpatients in an acute and life-threatening phase of the illness, so the clinicians could be more focused on the resolution of the organic issues than on other aspects not immediately related to the eating disorder. Finally, it is known that post-traumatic symptoms often maintain eating-related symptoms, causing an entangled situation: this mechanism could make eating-related symptoms more severe, putting the post-traumatic pathology in the background. However, it is recommended to pay more attention to post-traumatic symptomatology, especially in light of those studies describing a high prevalence of traumatic events in patients with AN, especially during childhood [13,42,43], and a negative treatment outcome in patients suffering from both eating-related and post-traumatic psychopathology [44].

The second part of the results suggests that SCID-based diagnostic assessments yielded a more fine-grained evaluation of the patients. Taking into account the researchers’ assessments, indeed, patients with comorbidity showed more severe results for all eating-related and general psychopathology symptoms compared to those without comorbidity. This is in line with studies reporting an exacerbation of symptoms when a psychiatric comorbidity is present [6]. In contrast, few differences emerged comparing patients with and without comorbidity according to the clinicians. This could be explained, again, by the propensity of clinicians to correlate certain symptoms to the eating disorder rather than to another diagnosis. From the clinical perspective, the severity of AN can be the “ace wins all”.

## 5. Conclusions

Overall, the present study focused on the importance of correctly investigating the presence of psychiatric comorbidities in AN. However, the diagnostic process should be conducted with caution, since artificially splitting a complex clinical condition into several pieces may prevent a holistic approach to the individual, encouraging unwarranted polypharmacy, and may represent a new source of diagnostic unreliability because clinicians may focus their attention on one or other of the different ‘pieces’, especially in those clinical contexts in which the coding of only one diagnosis is allowed. On the other hand, some comorbidities, especially anxiety disorders and PTSD, could maintain eating-related symptoms, so it could be useful to recognize the presence of such comorbid disorders and to specifically target them in the treatment. Moreover, an initial clinical evaluation of the patient also aims to create an alliance with the subject affected by AN; the alliance could be more difficult to achieve with the SCID-5 because of the highly standardized and strict structure. The solution could be in the middle of these two extreme pictures, and a higher focus on certain diagnoses is surely recommended.

Despite several strengths, including the sample size and the blindness of clinicians (i.e., clinicians were not informed that their diagnoses, routinely part of the hospitalization, would be compared to researchers’ diagnoses until the end of the present study), some limitations should be declared: Firstly, the study has a cross-sectional design preventing it from precisely establishing the timing of the comorbidities in relationship to the onset of AN; secondly, the use of SCID skip rules [45] meant that the research diagnostic process did not assess every diagnostic criterion in every participant, and more comprehensive clinical and research assessment may have uncovered additional sources of discordance; thirdly, some demographic data on participants (i.e., culture background, measures of income, education or socio-economic status) are not available; and, finally, we addressed just current diagnoses, and thus the influence of lifetime diagnoses on the concordance levels has not been considered. Despite this, the study has clinical implications, suggesting the importance of comorbidity assessment in AN, as well as theoretical relevance, adding knowledge to a field in which the debate is still open.

## Figures and Tables

**Table 1 jcm-11-07419-t001:** Concordance between clinical and SCID-5 diagnosis in the total sample of patients with AN.

	Clinical Diagnosis (%)	SCID-5 Diagnosis (%)	Cohen’s k (SE)	Interpretation of Agreement	Interpretation of Reproducibility
Major Depressive Disorder	30.3	34.4	0.197 (0.92)	poor	marginal
Persistent Depressive Disorder	0.8	10.7	−0.015 (0.015)	poor	marginal
Bipolar I Disorder	0	0.8	n.c.	-	-
Bipolar II Disorder	0	1.6	n.c.	-	-
Panic Disorder	0.8	20.5	−0.016 (0.016)	poor	marginal
Social Anxiety Disorder	1.6	9	0.130 (0.131)	poor	marginal
Generalized Anxiety Disorder	0.8	32	−0.016 (0.016)	poor	marginal
Agoraphobia	0	4.9	n.c.	-	-
Obsessive-Compulsive Disorder	1.6	2.5	0.388 (0.280)	fair	marginal
Post-traumatic Stress Disorder	0	17.2	n.c.	-	-
Substance Use Disorders	2.5	4.1	0.484 (0.220)	moderate	good
Mood Disorders *	30.3	47.5	0.264 (0.082)	fair	marginal
Anxiety Disorders **	2.5	46.7	0.021 (0.031)	poor	marginal

Cohen’s k values and agreement interpretation: k < 0 = no agreement, the agreement could only be attributed to the case; 0 < k < 0.2 = poor agreement; 0.2 < k < 0.4 = fair agreement; 0.4 < k < 0.6 = moderate agreement; 0.6 < k < 0.8 = substantial agreement; 0.8 < k < 1 = almost perfect agreement (Tyler and Birmingham, 2001). n.c. = not calculable. * patients with at least one mood disorder among major depressive disorder, persistent depressive disorder, bipolar I and II disorders. ** patients with at least one anxiety disorder among panic disorder, social anxiety disorder, generalized anxiety disorder and agoraphobia.

**Table 2 jcm-11-07419-t002:** Differences between patients with and without comorbidity according to SCID-5.

	Comorbidity (*n* = 85)	No Comorbidity (*n* = 37)	Test Statistics		
	Mean (SD)	Mean (SD)	t	*p*	Cohen’s d
Age, years	25.1 (9.4)	25.6 (9.1)	0.303	0.762	0.050
Years of illness	6.1 (6.3)	6.6 (8.1)	0.347	0.079	0.068
BMI	15.4 (2.5)	14.4 (2.9)	−1.856	0.066	0.038
EDE-Q					
Restraint	3.8 (1.9)	2.2 (1.9)	−0.357	<0.001	0.851
eating concerns	3.4 (1.4)	2.2 (1.3)	−0.3.939	<0.001	0.880
shape concerns	4.5 (1.4)	3.2 (1.4)	−3.606	<0.001	0.940
weight concerns	4.1 (1.7)	2.5 (1.7)	−3.828	<0.001	0.950
global score	4.0 (1.4)	2.6 (1.4)	−4.294	<0.001	1.010
STAI-state	59.3 (13.4)	49.8 (10.6)	−3.120	0.003	0.760
STAI-trait	62.1 (10.6)	47.3 (17.9)	−4.564	<0.001	1.130
BDI	18.5 (7.9)	13.3 (12.7)	−2.191	0.032	0.550

BMI = Body Mass Index; EDE-Q = Eating Disorder Examination Questionnaire; STAI = State-Trait Anxiety Inventory; BDI = Beck Depression Inventory. Cohen’s d values: negligible effect: d = −0.15–0.15; small effect: d = 0.15–0.40; medium effect: d = 0.40–0.75; large effect: d = 0.75–1.10; very large effect: d = 1.10–1.45; huge effect: d = 1.45.

**Table 3 jcm-11-07419-t003:** Differences between patients with and without comorbidity according to clinical evaluation.

	Comorbidity (*n* = 24)	No Comorbidity (*n* = 54)	Test Statistics		
	Mean (SD)	Mean (SD)	z	*p*	Cohen’s d
Age	28.2 (12.2)	23.5 (6.7)	−1.141	0.254	0.440
Years of illness	7.3 (7.5)	5.7 (6.4)	−1.246	0.213	0.226
BMI	14.9 (2.3)	15.2 (2.9)	0.318	0.751	0.116
EDE-Q restraint	3.8 (1.9)	3.1 (2.0)	−1.248	0.212	0.336
EDE-Q eating concerns	3.6 (1.7)	2.8 (1.4)	−2.521	0.012	0.554
EDE-Q shape concerns	4.4 (1.6)	3.9 (1.5)	−1.252	0.210	0.305
EDE-Q weight concerns	4.1 (1.6)	3.3 (1.9)	−1.626	0.104	0.422
EDE-q global score	3.9 (1.6)	3.3 (1.5)	−1.770	0.077	0.397
STAI-state	61.6 (11.7)	53.7 (13.2)	−2.261	0.024	0.611
STAI-trait	60.6 (10.1)	55.6 (16.7)	−1.164	0.244	0.400
BDI	20.0 (7.7)	15.6 (10.4)	−2.486	0.013	0.520

BMI = Body Mass Index; EDE-Q = Eating Disorder Examination Questionnaire; STAI = State-Trait Anxiety Inventory; BDI = Beck Depression Inventory. Cohen’s d values: negligible effect: d = −0.15–0.15; small effect: d = 0.15–0.40; medium effect: d = 0.40–0.75; large effect: d = 0.75–1.10; very large effect: d = 1.10–1.45; huge effect: d = 1.45.

## Data Availability

The data that support the findings of the study are available from the corresponding author upon reasonable request.

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
