# Peer review of "Diagnostic Concordance between Research and Clinical-Based Assessments of Psychiatric Comorbidity in Anorexia Nervosa"

_jcm, 2022, doi:10.3390/jcm11247419_

Round 1
Reviewer 1 Report
This is a generally well-written paper on an interesting (but not entirely new) topic.
I have some major and minor suggestions:
The title is misleading, since this methodology does not compare researchers and clinicians per se but rather clinical and research-based assessment of mental comorbidity.
I am not so sure if inter-rater reliability is the correct term. This term is mostly used to compare the results of two or more raters using the same instrument. I would suggest to rather apply the term “concordance” like in the title. In this line, it would be interesting to get information about the interrater-reliability of the researchers conducting the SCID-5.
ICD and DSM both use the term “mental disorders” rather than “psychiatric disorders”.
The authors should broaden the literature review on what is already known about the agreement between diagnoses made from clinical evaluations and standardized diagnostic interviews. I cannot find the meta-analysis published by Rettew et al 2009 or a similar study conducted in bariatric surgery candidates published by Mitchell et al. in 2010.
It is well known that structured interviews yield more diagnoses than clinical evaluations. However, that does not mean that this method is preferable and helpful for clinical use or treatment planning. State of the art might be the so-called LEAD standard as an operationalization of the best estimate diagnoses (Spitzer et al. 1983). A single clinical assessment (“at hospital admission”) might simply be not comprehensive enough to assess all problems of a patient. A first interview on admission also has the goal to establish a good therapeutic relationship and not only to correctly assess all comorbidities. During in- or outpatient treatment more mental disorders might emerge that are not detected during a single intake interview. It would be much more interesting to compare the discharge diagnoses with the SCID-5 diagnoses. This would be a more logical comparison.
Were only current diagnoses assessed? What about lifetime diagnoses? Usually, the concordance regarding lifetime disorders is higher compared to current diagnoses.
In Table 1 the authors might want to slightly modify the columns into:
· Only clinical interview
· Only SCID
· Clinical interview and SCID
Could you please explain what is meant with Mood Disorders and Anxiety Disorder under the heading Substance Use Disorders in Table 1?
Line 238: This “datum”? Probably this “result”?
Line 258: I am not sure if this statement is correct. On the contrary, OCD seems to be more prevalent in patients with AN compared to BN.
Drakes DH, Fawcett EJ, Rose JP, Carter-Major JC, Fawcett JM. Comorbid obsessive-compulsive disorder in individuals with eating disorders: An epidemiological meta-analysis. J Psychiatr Res. 2021;141:176-191
Line 282: There are longitudinal studies on the comorbidity between depression and eating disorders and the sequence of onset available.
Puccio F, Fuller-Tyszkiewicz M, Ong D, Krug I. A systematic review and meta-analysis on the longitudinal relationship between eating pathology and depression. Int J Eat Disord. 2016;49(5):439-54.
Line 310: Trauma and AN: Meta-analyses Caslini M, Bartoli F, Crocamo C, Dakanalis A, Clerici M, Carrà G. Disentangling the Association Between Child Abuse and Eating Disorders: A Systematic Review and Meta-Analysis. Psychosom Med. 2016 ;78(1):79-90
Lines 272 and 325: “full-blown diagnosis”?
I am not so sure about the clinical implications (line 347). Would the authors suggest that treatment planning or outcome is improved/positively influenced by assessing comorbid mental disorders with a SCID-5 interview? Is there any evidence supporting this notion? They correctly state that this might also be associated with potential problems such as polypharmacy and ”artificially splitting a complex clinical condition into several pieces”.
Author Response
As an overall statement, we would like to thank the reviewers for their fruitful and thoughtful comments which we believe improved our manuscript.
REVIEWER #1
This is a generally well-written paper on an interesting (but not entirely new) topic.
I have some major and minor suggestions:
The title is misleading, since this methodology does not compare researchers and clinicians per se but rather clinical and research-based assessment of mental comorbidity.
Thank you for the comment, you are right, we modified the title as you suggested, as we think it sounds now more precise. Now the title is: “Diagnostic concordance between research and clinical-based assessments of psychiatric comorbidity in anorexia nervosa”
I am not so sure if inter-rater reliability is the correct term. This term is mostly used to compare the results of two or more raters using the same instrument. I would suggest to rather apply the term “concordance” like in the title. In this line, it would be interesting to get information about the interrater-reliability of the researchers conducting the SCID-5.
Thank you for the comment. We used the term inter-rater reliability to be as coherent as possible with the statistics method adopted. However, you are right and we followed your suggestion replacing the term. As regards the literature about inter-rater reliability between different researchers adopting SCID-5, it would be really interesting to add this piece of information; however, at the best of our ability, we found just studies on validation of SCID-5 in languages different from English, and on the overall psychometric properties of SCID-5. Both types of studies are beyond the purpose of our research.
ICD and DSM both use the term “mental disorders” rather than “psychiatric disorders”.
Thanks, we replaced the term psychiatric disorders with mental disorders.
The authors should broaden the literature review on what is already known about the agreement between diagnoses made from clinical evaluations and standardized diagnostic interviews. I cannot find the meta-analysis published by Rettew et al 2009 or a similar study conducted in bariatric surgery candidates published by Mitchell et al. in 2010.
Thank you very much for the suggestion, we added these two important contributions to the introduction and discussion (lines 90-93, 252-254, and 279-284).
It is well known that structured interviews yield more diagnoses than clinical evaluations. However, that does not mean that this method is preferable and helpful for clinical use or treatment planning. State of the art might be the so-called LEAD standard as an operationalization of the best estimate diagnoses (Spitzer et al. 1983). A single clinical assessment (“at hospital admission”) might simply be not comprehensive enough to assess all problems of a patient. A first interview on admission also has the goal to establish a good therapeutic relationship and not only to correctly assess all comorbidities. During in- or outpatient treatment more mental disorders might emerge that are not detected during a single intake interview. It would be much more interesting to compare the discharge diagnoses with the SCID-5 diagnoses. This would be a more logical comparison.
We thank you very much for your rightful comment. The comment captures a crucial aspect of the diagnostic evaluation. However, our research study does not perform a single admission assessment, but a specific search done by clinicians for comorbidity. The text in the first version read "As regards the assessment of psychiatric comorbidity, two experienced psychiatrists (two of four psychiatrists in the clinical team) assessed for the presence of additional psychiatric diagnoses." and has been modified as follows for clarity: "As regards the assessment of psychiatric comorbidity, two experienced psychiatrists (two of four psychiatrists in the clinical team) assessed for the presence of additional psychiatric diagnoses, performing one or more additional interviews in the first 4 days of hospitalization” (lines 132-135). We also added to the discussion the suggested comment that the clinical evaluation, also aiming to create an alliance with the subject affected by AN, may be less thorough than the interview for the SCID (lines 370-373: “Moreover, an initial clinical evaluation of the patient also aims to create an alliance with the subject affected by AN; the alliance could be more difficult to achieve with the SCID-5 because of the highly standardized and strict structure”). Lastly, although we think that comparing diagnoses at discharge could be a good method, in this study the patients remain stable in the presentation of psychiatric comorbidity: we did a quick check on the discharge diagnoses. Moreover, we decided to compare SCID-5 diagnosis with clinical at-admission diagnosis for a matter of timing: indeed, the SCID-5 is administered during the first days of hospitalization, as stated in the procedure section.
Were only current diagnoses assessed? What about lifetime diagnoses? Usually, the concordance regarding lifetime disorders is higher compared to current diagnoses.
This is an interesting argument, thank you; we decided to address just current diagnoses for a matter of clarity; however, we added this to the limitations paragraph (lines 384-386: “finally, we addressed just current diagnoses, thus the influence of life-time diagnoses on concordance levels has not been considered”).
In Table 1 the authors might want to slightly modify the columns into:
- Only clinical interview
- Only SCID
- Clinical interview and SCID
The reviewer's suggestion leaves us with some doubts. We have preferred to keep the original manuscript version because it seems clear enough. We are open to further changes, should the reviewer deem it appropriate.
Could you please explain what is meant with Mood Disorders and Anxiety Disorder under the heading Substance Use Disorders in Table 1?
With these terms, we refer to the percentage of patients diagnosed with at least one mood or anxiety disorder, and we took into account them to calculate the concordance between the raters on the two categories as well as on the single diagnoses. We explained the concept in the text for mood disorders, but thanks to your comment we realized we had not specified that for anxiety disorders. Now we added the explanation for both categories in the legend of table 1, thank you.
Line 238: This “datum”? Probably this “result”?
Amended, thank you.
Line 258: I am not sure if this statement is correct. On the contrary, OCD seems to be more prevalent in patients with AN compared to BN.
Drakes DH, Fawcett EJ, Rose JP, Carter-Major JC, Fawcett JM. Comorbid obsessive-compulsive disorder in individuals with eating disorders: An epidemiological meta-analysis. J Psychiatr Res. 2021;141:176-191
Thank you for the comment. The review reports the highest prevalence of OCD in binge-purging AN, we amended it (lines 274-278: “Relatedly, our sample included just patients with AN, especially of the restricter subtype, while literature reports that patients suffering from the bulimic variant of AN are more likely to develop comorbidities, especially OCD”).
Line 282: There are longitudinal studies on the comorbidity between depression and eating disorders and the sequence of onset available.
Puccio F, Fuller-Tyszkiewicz M, Ong D, Krug I. A systematic review and meta-analysis on the longitudinal relationship between eating pathology and depression. Int J Eat Disord. 2016;49(5):439-54.
Thank you for the suggestion, we reported some longitudinal studies on the topic in the discussion (lines 308-313: “Longitudinal data could help to disentangle this issue; regarding depression, many longitudinal studies are available describing depression and eating pathology as concurrent risk factors [39]; other studies showed major depression disorder in childhood as risk factors for eating disorders [40], and the absence of depression as a positive prognostic factor [41]. However, longitudinal researches specific to AN are still lacking”).
Line 310: Trauma and AN: Meta-analyses Caslini M, Bartoli F, Crocamo C, Dakanalis A, Clerici M, Carrà G. Disentangling the Association Between Child Abuse and Eating Disorders: A Systematic Review and Meta-Analysis. Psychosom Med. 2016 ;78(1):79-90
We added this interesting review to the reference, thank you for the suggestion (line 340).
Lines 272 and 325: “full-blown diagnosis”?
With the term we meant a formal and independent diagnosis; however, we deleted it to enhance clarity and to follow your suggestion, thank you.
I am not so sure about the clinical implications (line 347). Would the authors suggest that treatment planning or outcome is improved/positively influenced by assessing comorbid mental disorders with a SCID-5 interview? Is there any evidence supporting this notion? They correctly state that this might also be associated with potential problems such as polypharmacy and ”artificially splitting a complex clinical condition into several pieces”.
In the clinical implications we tried to suggest, basing on the data, to adopt an “in-between solution” since both the SCID-5 and the clinical approaches could potentially lead to the negative consequences described. We tried to make clearer the concept (lines 373-374: “The solution could be in the middle of these two extreme pictures, and a higher focus on certain diagnoses is surely recommended”).

Reviewer 2 Report
Thank you for the opportunity to review this informative piece. It was interesting to learn about the effects that whole patient-based care and clinical judgement play on patient diagnosis.
A few highlights/comments below:
Line 19 - Spell out SCID with first use. It looks like first use is on line 14.
Line 29 - Consider rephrasing "characterized by high taxes". Perhaps, "characterized by high rates", if accurate?
Lines 92-102 - These seem to be better suited for Methods section. Consider consolidating into line 90-91 to abbreviate the introduction somewhat.
Line 115 - Why did you exclude those medical diagnoses?
Line 135 - Should this state "takes about 45 - 90" instead?
Line 176 - Change to, "The majority of the sample were female"
Line 180- Should benzodiazepine be plural?
Line 257 - Did you exclude patients with bulimia nervosa? If so, please update methods. If not, consider rephrasing and explaining your reason for not doing as others did.
Overall, there are redundancies between Results and Discussion section. Consider reducing content to be more succinct (no more than 4-5 paragraphs) and omit any phrases that restate information included in Methods and/or Results.
Is it possible that the Limitations and Conclusion headers were inadvertently omitted?
Author Response
As an overall statement, we would like to thank the reviewers for their fruitful and thoughtful comments which we believe improved our manuscript.
REVIEWER #2
Thank you for the opportunity to review this informative piece. It was interesting to learn about the effects that whole patient-based care and clinical judgement play on patient diagnosis.
A few highlights/comments below:
Line 19 - Spell out SCID with first use. It looks like first use is on line 14.
Thank you, done it.
Line 29 - Consider rephrasing "characterized by high taxes". Perhaps, "characterized by high rates", if accurate?
Thank you for the suggestion, we agree that the term could be confusing, so we deleted it; now we think the phrase sounds better (lines 30-31: “Anorexia nervosa (AN) is a severe mental disorder characterized by high mortality and psychiatric comorbidity”).
Lines 92-102 - These seem to be better suited for Methods section. Consider consolidating into line 90-91 to abbreviate the introduction somewhat.
We agree with you, thus we decided to delete part of these phrases and we tried to abbreviate the introduction, thank you.
Line 115 - Why did you exclude those medical diagnoses?
We excluded these and other medical conditions since potentially they could cause psychiatric symptoms not dependent on a mental disorder, but secondary to other neurological or systemic dysfunction. This is particularly true for neurological impairments (e.g., traumatic brain injuries or stroke impacting frontal lobes could give behavioral changes and symptoms as well as mood deflection or flattening): so we think that excluding these conditions is a good strategy to minimize biases in the diagnostic process.
Line 135 - Should this state "takes about 45 - 90" instead?
Amended, thank you.
Line 176 - Change to, "The majority of the sample were female"
Thank you for the comment. We rephrased in a clearer way (line 183-184: “Female patients represented the majority of the sample (98.4%)”.
Line 180- Should benzodiazepine be plural?
Amended, thank you.
Line 257 - Did you exclude patients with bulimia nervosa? If so, please update methods. If not, consider rephrasing and explaining your reason for not doing as others did.
Thank you for the interesting question. We thought a lot about it, but we decided to actively recruit patients with AN for this study because the vast majority of patients arriving at our centre suffer from AN; thus including patients with bulimia nervosa could have resulted in a small subgroup with consistent phenotypical differences potentially influencing the assessment of comorbidity. We listed the diagnosis of AN as an inclusion criterion of the study. Moreover, it is necessary to bear in mind that the functioning of the Italian health system provides hospitalization especially for subjects suffering from AN, while subjects suffering from BN receive therapies in DH or intensive outpatient.
Overall, there are redundancies between Results and Discussion sections. Consider reducing content to be more succinct (no more than 4-5 paragraphs) and omit any phrases that restate information included in Methods and/or Results.
Thank you for the suggestion, we did our best to delete redundancies and rephrase the text to enhance clarity.
Is it possible that the Limitations and Conclusion headers were inadvertently omitted?
We had decided to write a single final section; however your comment made us think that dividing the discussion and conclusion could be clearer, so we add the header, thank you.

Round 2
Reviewer 1 Report
The authors did a good job in revising the manuscript.
Reviewer 2 Report
Thank you for the prompt edits. I believe this will add to scientific knowledge.